# A thematic analysis of flu vaccine hesitance in ethnically minoritised communities in Liverpool

Anna Powell *, Marie Claire Van Hout ¤a, Deborah Connors ¤b, Catharine Montgomery

School of Psychology, James Parsons Building, Liverpool John Moores University, Liverpool, United Kingdom

¤a Current address: South East Technological University, Ireland
¤b Current address: The King's Trust, Nottingham, United Kingdom
* A.Powell1@ljmu.ac.uk

## Abstract

### Background

Seasonal influenza causes around 15,000 deaths yearly in the United Kingdom. Low vaccine uptake is more prominent in ethnically minoritised communities and deprived areas, leading to poorer outcomes.

### Aims

To understand influenza vaccine hesitance in ethnically minoritised communities in Liverpool from multistakeholder perspectives.

### Methods

Semi-structured interviews and focus groups were conducted with members of the public (n = 55), community engagement workers (n = 14), primary healthcare staff (n = 20), and policy professionals (n = 10). Data were analysed thematically.

### Results

Six themes were identified. *Beliefs* about vaccine safety, necessity, and efficacy often arose from misinformation, misunderstanding, or negative experience. *Trust* in vaccine information depended on source familiarity, credibility, and perceived intentions, while trust in the healthcare system had decreased due to cultural and COVID-19 concerns. *Accessibility* of accurate vaccine information was poor, due to language and literacy barriers. *Community* opinions/experience shaped perceptions, while community organisations were trusted but needed resources/stability. *Healthcare* staff described low morale, time/resource constraints, and uncertainty in addressing cultural concerns. Ultimately, *Alliance* indicated a desire for better integration between healthcare and communities, particularly for developing/distributing accurate, culturally relevant, and accessible information.

**Data availability statement:** Data cannot be shared publicly as the Institutional Ethics Research Committee (LJMUREC) approved the study on the basis that anonymised transcripts will be deposited on a restricted basis on the LJMU open access data repository. Therefore, anonymised transcripts will be available upon reasonable request via https://opendata.ljmu.ac.uk/ for researchers who meet the criteria for access to this data (e.g. academic or health care researchers working in this field).

**Funding:** This study was funded by The Pandemic Institute, through partnership with CSL Seqirus UK Limited [TPI-SEQ3-CM]. The funders had no role in study design, data collection and analysis, decision to publish, or preparation of the manuscript.

**Competing interests:** This study was funded by The Pandemic Institute, through partnership with CSL Seqirus UK Limited, an influenza vaccine manufacturer. The funder was not involved in the study design; in the collection, analysis or interpretation of data; in the writing of the article; or in the decision to submit it for publication, all of which were executed with no input from them.

## Conclusion

To address influenza vaccine hesitance, stakeholders should collaborate to improve access to reliable information (to support development of pro-vaccine beliefs) via tailored communication and culturally informed training for healthcare staff; aim to increase trust by, for example, ensuring access to familiar staff and employing community members; and foster alliance via long-term support of community organisations through funding, accurate information, and training.

## Introduction

Seasonal influenza causes between 290,000 and 650,000 deaths globally per year [1], and puts significant strain on the United Kingdom (UK) healthcare system [2]. During the 2024−25 UK flu season, there was a 3.5 times higher daily average of people (5,408 patients) in hospital with influenza, leading to over 20 NHS trusts declaring critical incidents, with one hospital in Liverpool reporting emergency care waits of up to 50 hours [3]. The influenza disease burden is felt across the healthcare system, with GP consultations, hospitalisation rates and ICU/HDI admissions all higher during the 2024−25 season than in 2023−24, with influenza-attributable excess mortality estimated at 7,757 deaths (compared to 3,555 deaths in 2023−24) [4]. Moreover Liverpool, the city where this study was conducted, has the 10th highest mortality rate from influenza or pneumonia, of local authority areas in England (54.1, compared to 34.2, people per 100,000 England average) [5]. UK hospital admission statistics show that minority ethnic groups have higher disease severity with the Pakistani ethnic group having the highest hospital admission rates for influenza (2.7 times higher than ethnically White group), and the Black, African, Caribbean, or Black British ethnic group having 1.6 times higher hospitalisation rates than people who are ethnically White [6].

Vaccination against influenza effectively reduces morbidity and mortality [7–10]. Estimates by the International Longevity Centre UK, suggest that influenza vaccination prevents between 200,000–600,000 cases and 6,000–10,000 premature deaths yearly in England [8]. In the UK, influenza vaccination is offered for free to eligible individuals every year, with vaccination programmes starting in September/October and running until March. Eligible individuals are those who meet at least one of the following criteria: aged 65+, clinically at risk, pregnancy, aged 2−3 years, children in school years reception – year 11, people in long-term residential care homes, carers of older or disabled people, frontline health and social care staff and household contacts of people who are immunocompromised [11]. However, vaccination rates remain low [12]. Indeed, compared to the recommended 75% Vaccination Coverage Rate (VCR), the UK only achieved 65% VCR in the 2021−2022 season [13]. More recently, in January of the 2024−2025 season, vaccination rates were lower than the previous year indicating a potential worsening of the scenario for controlling infection [3]. In the UK, data from the 2024−25 flu season shows that while there are some variations in rates of vaccination in various minority ethnic groups, for most eligibility



criteria, White British ethnicity has significantly higher rates of influenza vaccination than all other ethnic groups [14], with the exception of pregnant women and preschool children where Chinese ethnicity had the highest rates. For example, in those aged over 65 years, the White British ethnicity had 4.2 percentage points higher vaccination rates than average at 74.9%, with lowest uptake (ranging from 40.7% to 49.0%) in the Black and Mixed Black groups. Similar patterns are seen globally where minority ethnic groups have lower vaccination rates; for example in the USA, Hispanic, Black and Native American ethnicity have lower flu vaccination rates than ethnically White groups [15]. The lower rates of vaccination in minority ethnic groups are a concern as influenza outcomes are disproportionately poorer in certain groups, including ethnically minoritised communities [15,16]. To improve outcomes and address low uptake requires examination of 'vaccine hesitance' – refusal or delay in acceptance of vaccination [17]. Systematic reviews of the general population indicate various reasons for influenza vaccine hesitance, such as worries about safety/effectiveness, low trust in healthcare, lack of information or misinformation, and low perceived risk of influenza [12,18,19], as well as cultural concerns, including of halal status of vaccines, impact on fertility, racial fairness, prior experience of discriminatory behaviour, preference for natural remedies or religious practices [20], and accessibility [19]. Another systematic review [21] which focused on ethnically minoritized groups but was not specific to influenza, found several barriers that overlap, including mis/lack of information, perceptions of low risk or of vaccines being ineffective/harmful, low trust of healthcare system, and accessibility of vaccines.

However, research suggests that vaccination barriers may be specific to the specific community, with migrants more likely to report language/communication barriers, while Black ethnic groups reported concerns around vaccine safety and trust of healthcare systems [22]. Furthermore, inter and intra-area differences in vaccine uptake (and therefore, hesitance), are not always consistent, with two studies in England finding contrastingly that Black patients are either more [23] or less [24] likely to receive the influenza vaccine than other groups. It is possible this could be due to complex interactions between sociodemographic factors (intersectionality), such as ethnicity, age and disability [25]. Indeed, UK studies show that impact of ethnicity and sex on uptake differs across age [24], as does the impact of deprivation [23,24]. One UK study found that inequalities in influenza vaccine uptake between ethnic groups were greater in more income-deprived areas, and in women, reflecting intersectionality that exacerbates health inequalities [26]. Therefore, considering barriers and facilitators to vaccination across/between areas could be misleading, and it may be more useful to understand vaccine hesitance from the perspective of smaller, localised, income-deprived communities, to design targeted intervention.

Liverpool is the third most deprived local authority in England and Wales [27]. It is a diverse city, in which 22.7% of the population identified as non-White English, Welsh, Scottish, Northern Irish, or British, in the UK 2021 Census [28]. It has the 11th largest deprivation gap between neighbourhoods (50.4 percentage points between the least and most deprived; Office for National Statistics [27]), so likely has large internal disparities. Vaccination rates in Liverpool are lower than the national averages, particularly in ethnic minority communities [29,30], a pattern which is seen in similar cities in the North of England such as Manchester [26], and more diverse and larger cities in the South of England such as London [31]. Health inequalities in influenza outcomes result in marginalised communities being disproportionately affected by vaccine hesitance [32,33]. So, as with other illnesses, an increase in uptake of vaccination may have disproportionate benefits in this context, where disease burden is higher [34].

Additionally, amongst vaccines, influenza is unique for several reasons, as it requires repeated, seasonal vaccination at multiple life stages, and fluctuates yearly in both effectiveness and recommended risk groups, which may influence public understanding of vaccine benefits and influenza risk. Finally, influenza vaccination has shown to be linked to the recent COVID-19 pandemic, as influenza vaccine uptake associates with COVID-19 vaccine attitudes, and people are more likely to experience vaccine fatigue (Skyles et al., 2023), perhaps in part because both are infectious respiratory diseases. Therefore, prior literature suggests potentially unique current barriers regarding influenza vaccine acceptance, which are important to understand in ethnically minoritised communities, particularly within localised, deprived areas, such as Liverpool. This insight can then be examined for applicability to areas with similar deprivation and ethnicity profiles. The current



study aims to qualitatively understand influenza vaccine hesitance in the Liverpool region, with the intention being to guide targeted intervention.

## Method

### Participants and recruitment

This study used a qualitative design. Four groups were recruited from Liverpool, UK:

i) adult (aged 18+) members of the public who self-identified as being part of an ethnically minoritised group and eligible for a seasonal influenza vaccine (or had an eligible child),

ii) community engagement workers,

iii) primary healthcare staff,

iv) policy professionals (such as service commissioners and policymakers).

Potential participants were identified via various networks, including Primary Care Networks, Cheshire & Merseyside ICB, NIHR Applied Research Collaboration North West Coast and the Clinical Research Network, Liverpool City Council, and community organisations using purposive sampling.

Participants were recruited purposively across the four target groups. Policy professionals, healthcare staff, and community engagement workers were identified through existing collaborative links and face-to-face interactions at Liverpool City Council learning events, and were invited via email, newsletters, or direct conversation. Some members of the public were identified and approached by community organisations acting as gatekeepers, who translated and disseminated study information through in-person conversations, phone calls, and direct social media messaging. These organisations also supported participation by setting up Teams/Zoom links for remote meetings, providing rooms for in-person meetings, and arranging interpreters who spoke local dialects where needed. Study information was also provided in an easy-read format for community organisations. While many participants were directly identified and invited, some were recruited through open calls (e.g., newsletters and indirect social media posts). As noted in the later described limitations, the use of community organisations as gatekeepers to recruit members of the public likely introduced a positive bias towards these organisations.

Data were collected from 19th January to 29th May 2024. Ninety-six people took part (see Table 1 for participant characteristics): 55 members of the public, 14 community engagement workers, 20 primary healthcare staff, and 10 policy professionals. Three were both primary healthcare staff and policy professionals. Most were female (see Table 1 for demographics). Sample size was guided by the concept of information power (preferred to 'data saturation'; Clarke and Braun [35]), which indicated that a larger sample was needed. The study aim was moderately focused, with purposive sampling used to ensure relevance and variation. While no established theory was applied, the approach was informed by existing literature. Interviews were conducted by researchers trained in qualitative methods and familiar with the topic, supporting high-quality dialogue. A descriptive cross-case analysis strategy, rather than case-by-case analysis, further justified a larger sample. Focus groups were conducted across 34 participants (with individuals from the same participant group), and consisted of between two and five people, while 62 individual interviews were conducted. Forty-one people not fluent in English were able to participate due to the support of community organisations.

### Ethical approval

This study was approved by the Health Research Authority (IRAS number: 334970) and the Liverpool John Moores University Research Ethics Committee (Reference: 23/PSY/081).



**Table 1. Demographic characteristics of participants.**

| | Mean | Range |
|---|---|---|
| Age | 46.02 | 21-83 |
| | **n** | **%** |
| Gender | | |
| Male | 18 | 18.75 |
| Female | 78 | 81.25 |
| Ethnicity | | |
| **Asian or Asian British** (Chinese, Chinese Hong-Kong, Hong Kongese, Pakistani, Indian) | 15 | 15.63 |
| **Black, Black British, Caribbean or African** (African, Black British, Sudanese, Moroccan, Libyan) | 6 | 6.25 |
| **Mixed or Multiple ethnic groups** (Mixed, Black Portuguese, British Bangladeshi, British Muslim, Caribbean Asian, White and Asian, White Mixed) | 8 | 8.33 |
| **White** (British, Polish, Roma, Roma Gypsy, Romani Gypsy, Romanian, Romanian Gypsy, Romanian Roma, White) | 25 | 26.04 |
| **White British** | 26 | 27.08 |
| **Other** (Arab, Arabic Yemeni, Afghans, Egyptian, Iraqi, Kuwaiti, Syrian, Yemeni) | 16 | 16.67 |

## Procedure

Participants were given information on the study and if they agreed to take part a suitable time was arranged with either AP or DC, with interpretation as required. Most sessions were conducted by AP, the Principal Investigator (n = 81), who holds a PhD in Psychology and Neuroscience and has 11 years of experience in both qualitative and quantitative research. DC conducted the remaining interviews, and had been trained in qualitative research as part of MSc Stage 1 Health Psychologist training. To ensure consistency and data quality, DC was further prepared by AP through a structured process: DC initially shadowed several interviews, then conducted one interview under AP's supervision, followed by a debrief and feedback. AP continued to monitor the quality of DC's recordings throughout the data collection period. The choice of interview or focus-group was context-dependent and related to preferences of participants, interpreters and/or community organisations. For in-person data collection, written consent forms were signed at this meeting, prior to data collection occurring. For online (Zoom or Teams) data collection, a signed written consent form was returned via email, prior to the meeting. At the start of each session, the study overview was described, including confidentiality, the right to withdraw, and giving the opportunity for questions. Following the provision of informed consent, including to either audio- (in-person) or video- (online) record, sessions began with the collection of demographic data and then addressed the topics of interest. All sessions followed semi-structured schedules with open-ended questions, developed for each participant group by drawing on prior literature (see S1 File for topic guides). Participants received a shopping voucher with a value of 25 GBP (NIHR involve rate) as a thank you for their time.

## Data analysis

Recordings were transcribed using the MS Teams transcription function or Otter AI and were then anonymised and checked for accuracy by the research team. The data was then analysed inductively in NVivo (R14.23.3), following the six steps of thematic analysis [36]. Data familiarisation, initial coding, and searching for themes was undertaken by AP, with a subsection of the data (n = 14, 14.58%) also processed by DC, to ensure similar interpretations of the data were being reached. Reflexivity was maintained through discussions between AP, who is White British, and DC, who is Indian and a member of an ethnically minoritised community in the UK, ensuring diverse perspectives were brought to the analysis. The initial set of themes were subsequently discussed amongst the research team, and explored further against the data,

enabling them to be reviewed and refined until final themes were agreed upon, defined, and named. These final themes were then shared and agreed with the community organisations as a form of member checking, ensuring they reflected the conversations the organisations were witness to and involved in with their communities. Corresponding anonymised data can be made available upon reasonable request through the LJMU open access data repository via this link https://opendata.ljmu.ac.uk/.

## Results

Six themes were identified: 1) Beliefs, 2) Trust, 3) Accessibility, 4) Community, 5) Healthcare, and 6) Alliance. The connections between these themes as barriers to the flu vaccine in ethnically minoritised groups are displayed in Fig 1.

Several subthemes were also identified, see Table 2 for a summary of these and the settings in which they were understood to occur.

Of the themes, the most commonly occurring were Beliefs, Trust, and Accessibility. Within these dominant themes, certain sub-themes emerged as especially prevalent. For Beliefs, the most frequently recurring sub-theme was "*Is it safe, acceptable, or necessary?*". Under Trust, "*Sorry, do I know you?*" and "*Do you have my best interests at heart*" were the most prominent. Within Accessibility, "Un*Informed choices*" stood out as particularly significant. Themes and subthemes are discussed in more detail below, presented in an order that best supports a coherent and meaningful narrative, rather than reflecting frequency.

### 1. Beliefs

**1.1. Who am I?** There were three distinct vaccine attitudes amongst communities related to willingness to receive vaccine information. Broadly, people were either anti-vaccine and unwilling to engage, were hesitant and unsure but willing to engage, or were already pro-vaccine.

*"Nobody could convince her. No. Jesus' mother could come to her and say to do the vaccine, and she wouldn't do it."* (P84, Public)

Several people discussed experiencing vaccine fatigue, where despite having previously been pro-vaccine, they were now (since the COVID-19 pandemic) less interested in engaging in vaccine behaviour or conversation.

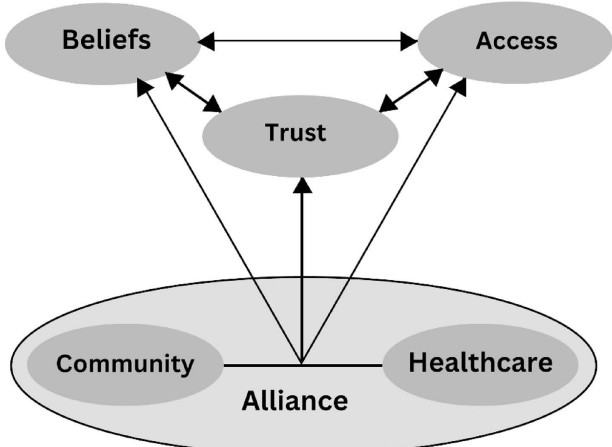

**Fig 1. Relationships between the main themes that act as context-dependent facilitators or barriers to the flu vaccine in ethnically minoritised groups.**

**Table 2. Themes and subthemes acting as contextual facilitators or barriers to flu vaccine in ethnically minoritised groups, and settings in which these occur.**

| Theme | Subtheme(s) | Summary | Location(s) |
|---|---|---|---|
| **Beliefs** | | | |
| | *Who am I?* | People either anti-vaccine, hesitant, or pro-vaccine. Influenced willingness to engage with vaccines/ discuss them. | Individual. |
| | *Is it safe, acceptable, or necessary?* | Viewed flu as not serious, and vaccine as unsafe or culturally irrelevant/ inacceptable. | Individual, Community, At home. |
| | *How do I usually access information?* | Health beliefs informed by many sources, including internet, social media, families, previous countries, and community organisations. | Online, TV, Messaging/Phone, Community, At home, GP surgery. |
| **Trust** | | | |
| | *Sorry, do I know you?* | Perceived credentials and/ or familiarity made sources more trusted. | Online, GP surgery TV, Messaging/Phone, Community, At home. |
| | *Do you have my best interests at heart?* | Reduced trust in authority and medical institutions – due to Racism, scandals, poor representation/ accessibility, and COVID-19 decisions. | GP surgery, Community, Online, TV, Messaging/Phone. |
| | *How well do I know myself?* | Disconnect of information sources thought trustworthy vs used. | Individual. |
| **Accessibility** | | | |
| | Un*Informed choices* | Language, literacy, and opportunity barriers reduced accessibility of accurate information. Religious worries unaddressed. Confusion/ fear around simultaneous promotion of COVID-19 vaccine. | GP Surgery, Online, TV, Messaging/Phone, Community. |
| | *Getting the vaccine is (mostly) easy* | Physical access mostly positive, except where poor access to a translator or to GP, or when unsure how to access. | GP surgery, Pharmacy, Online, Messaging/Phone. |
| | *But anything else is (mostly) hard* | Negatively viewed accessing GP for other reasons – feelings of disconnected agenda between patient & GP. | GP surgery. |
| **Community** | | | |
| | *Liverpool: Us and Them* | Strong valuing of free will and uniting against authority. Racism a complex current and past issue. | Community. |
| | *Close-knit* | Close communities with shared experiences and strong trust. Negative vaccine stories spread easily. | Community. |
| | *We are here* | Communities and trusted organisations communicate effectively. Positive work already happening, resources needed. | Community. |
| **Healthcare** | | Healthcare staff worried about lack of time, competition between services, resources, and how to address culturally specific concerns. | GP surgery, Community. |
| **Alliance** | | | |
| | *Current disconnect* | Between healthcare and the community. | GP surgery, Pharmacy, Community. |
| | *Co-production and sustainability are key* | Desire for flu messaging campaigns to be culturally informed, and for community organisations to be supported. | GP surgery, Pharmacy, Community, Online, Messaging/Phone. |

*"…with the COVID, there was so much vaccines you have to take … And then comes, come along, and you have to get the flu vaccine. And I thought I've had enough vaccines and boosters… How much vitamins you going to take (laughs), you know, how much oranges you're going to eat?... how much are we expected to do not to get flu or COVID again?"* (P26, Public)

**1.2. Is it safe, acceptable, or necessary?** The flu was not viewed as a serious risk by many members of the public and by some clinicians. Some individuals confused the flu with other conditions (e.g., common cold, COVID-19). It was generally considered as a low priority.

*"…you will often hear people say, "oh, I've got a bit of the flu erm, you know, I've got a bit of cold, oh I've got flu like symptoms". People, unless you've had flu, you don't know, you don't know how poorly you, how you can be."* (P1, Policy Professional)

Individuals also worried that the flu vaccine did not work or was unsafe. This was due to inconsistent information, misinformation, or a prior negative experience.

*"I decided what, if I have all this, all this vaccine from the COVID, should I be protected enough not to have the flu vaccine? …That's why I didn't have the vaccine last year."* (P26, Public)

*"…after that [flu vaccine] she got ill for 12 months, and uh she cough and cough … she recovered and after that she doesn't want to take any uh vaccination about this flu"* (P80, Public)

Some people felt that health or religious behaviours offered adequate flu protection while others were unsure regarding the vaccine's halal status. Some participants described fear of needles.

*"…for Chinese, we got so many different myths and beliefs and we got herbs and people, you know, like … gingers mixed with hot waters or ginseng, you know, like they … feel like that will help to protect themselves especially for the seasonal change"* (P16, Community Worker)

**1.3.  How do I usually access information?**  Different communities access (and share) information via different platforms and in different formats, with the internet, social media, and the media often playing a key role.

*"…the Chinese community, there's this, the social media called WeChat… and Polish community social media is great … where there's low literacy levels… the Irish Traveler community... having a trusted person going to speak to them alongside the professional … it's not a one size fits all"* (P7, Community Worker)

Many people looked to information from countries they had previously been living in or in which they had family still living.

*"…maybe there might be a health channel [YouTube] from other country, or China or Taiwan or Hong Kong that I can referring to, for those. Maybe part of it, I can, you know, apply on myself that I found is more culturally appropriate. I will listen, I will follow that advice or guidance."* (P39, Public)

Community organisations were often described as important and trusted health communicators, indicating positive use of social media for health information sharing, in appropriate formats.

*"Short videos, where someone is speaking to you, it works well with Roma community… they will quite often use WhatsApp to communicate by sending voice notes…community champions who work with Roma community…every time there is a message incoming from school, she will then send the voice note, telling them exactly the same thing that it says in the message."* (P10, Community Worker)

Communications via families and social circles, along with religious and other community leaders were also described as influential.

*"I know that when we did COVID, or when the team was doing COVID immunizations, at the mosque, as soon as they got the Imam on board, and was like, the gelatine, free gelatine, and all these other things the, honestly, the queue was round the block"* (P3, Community Worker)



## 2. Trust

### 2.1. Sorry, do I know you?
Both credentials and social/physical familiarity impacted trust. Family, friends, and other members of the community were described as well trusted, as were community leaders, community organisations, and familiar clinicians with whom a relationship had been built.

*"…she's saying that normally when that, one of your family, like somebody from your family is tells you that "this is good for you, this is going to support you and keep you safe", that makes you more stronger and become more believing on what your family says because you always gonna trust your family."* (P56, Public)

*"…if you know someone and then you have relationship, especially if you have that relationship over a long time, then kind of people will trust that you've got their interests at heart… they'll listen to you if you say," yeah, you should get it".*" (P11, Primary Care)

### 2.2. Do you have my best interests at heart?
For many, trust seemed to have eroded between the public and the healthcare system (considered another form of authority). This was due to various factors, including systemic and interpersonal racism…

*"Sims is still being taught in the midwifery course at [hospital name redacted]… a renowned medical racist, the guy's sat in a portrait… you can see it, the black body parts in, you know, jars and whatnot … there's an integrated, like, the practices and thought processes."* (P3, Community worker)

*"…the GP spoke to them loudly and slowly. And it's almost like they're undermining your ability to think… I think it's very damaging, to having relationship"* (P10, Community Worker)

…medical scandals or misinformation, some of which were specific to vaccination (e.g., MMR vaccine), or to ethnically minoritised communities…

*"So a lack of trust in healthcare services... various sort of medical scandals... But specifically for ethnic minority groups there is more… using patients for clinical trials without really fully being consented… I think there's the historical context that applies more to ethnic minority groups, and then that is overlaid by the general mistrust in society."* (P2, Policy Professional)

…poor representation of providers from ethnic minority backgrounds…

*"I also think there's a much bigger issue of better representation. I don't if that's really, that feels like that's not a real commitment to change, but certainly our staff [should] be more reflective of the communities that we want to serve."* (P2, Policy Professional)

… 'authority' decisions during the COVID-19 pandemic…

*"I think it [COVID-19] just wasn't handled well… So there was a lot of mistrust with the government umm… So you, the public, aren't gonna believe it, and in turn, I feel filters down in regards to Public Health England, in regards to NHSE"* (P21, Primary Care)

…low accessibility due to language, literacy barriers, and digital exclusion leading to feelings of low priority…

*"…we don't have great kind of front door access for people who don't speak English as the first language… how do you deal with the receptionist on the phone when you're trying to make an appointment? How do you deal with an online system that's all in English? So I think that's a an extra factor that means there's not a lot of opportunity to rebuild that trust really at practice level."* (P2, Policy Professional)

In addition, people had opposing views on whether the NHS is a trusted source of information.

*"Their [UKHSA] evidence shows… I think it's 93% of people want health care professional to give them information and they want something NHS branded"* (P15, Policy Professional)

*"Another thing is that we found was quite helpful through work through LSTM was to create content that doesn't have NHS branding on it. I know it sounds really bad. But people sometimes look at it, and they just don't want to hear it because they think it's a propaganda to government, pharmaceutical propaganda."* (P10, Community Worker)

**2.3. How well do I know myself?** People demonstrated a disconnect about which sources of health information they trusted or were influenced by. They described social media and family as causing them concerns about vaccines, but also as being not credible (family) or untrustworthy (social media).

*"I've had conversations with people and they say, "oh no, we don't trust the vaccine", because of the fake news and access to social media and YouTube and TikTok and stuff… Umm but I've, said to people but, "if you know you had chest pains and thought you had an heart attack, the first person you'd call this be NHS, you would phone for an ambulance so you don't trust what the NHS is saying about vaccinations, but you would trust them happily if you become seriously ill". So, it's that, it's incongruent."* (P5, Community Worker)

*"…he doesn't trust anything in their [YouTube] websites. It just that when mom sent him something [YouTube video] he just say he just watch it."* (P71, Public)

Similarly, in most contexts they described trusting advice from their GP, but they were less inclined to trust GP advice when it was regarding vaccines.

*"They're quite good anyway, [doctors], apart from the vaccine"* (P97, Public)

## 3. Accessibility

**3.1. Uninformed choices.** Language and literacy barriers meant that for many of the community, accessibility of accurate information was poor, potentially increasing their vulnerability to misinformation.

*"…sometimes they'll come somebody came to speak in English… and they don't understand anything. Should be the same language."* (P66, Public)

*"…she's been saying that she wants to see a video rather than letter because it's easier because she doesn't read and write."* (P55, Public)

People also felt they did not have the opportunity for conversations with trusted healthcare staff about the flu vaccine.

*"Imagine yourself like…you don't know what can be in there…if you're gonna go… "excuse me, what you have there, what you gonna put me? What you gonna inject me in this moment?", you cannot put them explain everything. They just do their job"* (P61, Public)

There were concerns about whether the vaccine was culturally acceptable, and while the injectable flu vaccine does not contain pork gelatine this was not always made clear.

*"…people also think about religions, because I think with flu in particular, there was some kind of rumours that it contained like pig fat or pork products."* (P14, Policy Professional)

*"…a lot of things that have not been really communicated, and then because people have these conversations between themselves. Then obviously that starts to create an uneasy atmosphere, they think, "but my doctor didn't tell me…"… So I think there's needs to be a more of a transparency"* (P10, Community Worker)

People were also generally more frightened of the COVID-19 vaccine, and worried that this would be used or added to their flu vaccine, potentially due to the two vaccines being targeted at the same groups during the winter vaccine campaigns. This impacted the language that community workers would use with some groups.

*"…the COVID-19 vaccinations, we always help to promote a flu jab as well, especially during the time and you know, like they can get the both jab together, you know? … A majority of people don't want to get this … we as a Chinese, we believe you know, like they got different effects. So, you don't want to be, to add it up to be one serious one."* (P16, Community Worker)

*"We don't, we don't use the [term] vaccination. We use [the term] injection only, because a lot of Eastern Europeans in our countries, for example in Romania, a lot of uh people they hear first time vaccine with the COVID… when they hear vaccine is equal with COVID-19"* (P13, Community Worker)

**3.2. Getting the vaccine is (mostly) easy.** Although healthcare staff expected accessing the vaccine to be a barrier…

*"…it's also about just kind of making it really easy for people so that they can … just get it without booking something or having to wait"* (P11, Primary Care)

…most members of the public described physical access of the vaccine positively…

*"I mean, although it is against my real advice, but getting the vaccine is quite easy."* (P45, Public)

*"…we can make an appointment at GP… so we can have a date and time of the vaccine. Or if it's more convenient, because we can choose so then we can make an appointment and have it done in the pharmacy."* (P75, Public)

…except for a few members of the public who experienced language barriers, did not know where to access it, or struggled with appointment flexibility and booking.

*"…the only problem is with language barrier … every time she's been asking for a translator they will say "no sorry, we don't have it available", and that makes her not going to access the appointment for the vaccine."* (P56, Public)

*"I'm really confused where to go?"* (P78, Public)

*"…one of the reason that I've been skipping this flu jab for some years, the reason why is because of the work commitment, time restraints, and also it's really hard to get into GP line to book an appointment."* (P39, Public)

**3.3. But anything else is (mostly) hard.** However, many people expressed negative views around accessing the GP when wanted by the patient, which led to feelings of the vaccine being prioritised. This in turn lowered trust and created a feeling of a disconnected agenda between patient and healthcare.

*"…they're so excited to give us the flu vaccine, but they're not excited about something else"* (P69, Public)

*"…saying things like "I can't get an appointment at the GP, but when they want me to go and have my vaccine, oh they're quite happy to text me and tell me I've got to come in and have my vaccine""* (P17, Community Worker)

## 4. Community

**4.1. Liverpool: Us and Them.** The identity of Liverpool came through strongly, with the perception that it is often united against authority and values "free will":

*"I think diverse communities and also a lot of people in Liverpool more generally is [do not trust], politicians and the government. I think that's fairly clear. I mean probably more so in Liverpool, I think everything feels a little bit more heated here."* (P17, Community Worker)

…but simultaneously as an area with current and historic poor treatment of its ethnic minorities.

*"I think Liverpool as a city has a real racism problem, I think any institution space that you go into, you will experience or there will be an element of racism within that, I think I honestly believe that it's literally within the foundations of this city. And unfortunately, you can't get away from it."* (P3, Community Worker)

**4.2. Close-knit.** There was a lot of shared experience within communities, important for understanding the needs of these communities and their vaccine attitudes. Negative information seemed to spread more easily, perhaps due to language acting as a barrier to other information.

*"…these communities are… very socialized to each other. So whenever they heard something is just like … "…I heard this thing from someone else"."* (P82, Community Worker)

*"It's a very tight community. I mean, they all come from the same village in Romania. Yeah. So they're cousins. Yeah, everybody is connected."* (PB, Public)

**4.3. We are here.** The community itself was best placed to communicate with, and provide access to, its own community. Community organisations and their staff were already trusted and knew how to communicate effectively with the community they serve and have access to local translators/interpreters who know the local dialect. These organisations are already supporting the health of their community:

*"I've got various, I mean, just 14 languages spoken within my team and 10 different cultures. Umm so they're best informed about how they can communicate with those communities, if that makes sense?"* (P7, Community worker)

*"[about volunteering] …we had this awareness session for the clients about bowel cancer. So before, I gave the talks, and before giving the talk, I just went to the bowel cancer [website], I did my research by my own self, I downloaded the leaflets and I tried to understand them in my own language as well, like, what does it mean and what suitable words I can found to in easy words…. there are people from they speaking Cantonese and … Urdu and Persian, and so some of the languages was like, I could speak so I was very, it was easy for me like to deliver. And the same time, we had a translator interpreter from the NHS for the Cantonese."* (P82, Community Worker)

However, these organisations needed resources and stability, to enact lasting change.

*"…too often, it's at the goodwill of people who genuinely care about their own communities, that they would volunteer... and it's not good enough."* (P5, Community worker)

### 5. Healthcare (staff concerns)

Healthcare staff consistently cited lack of time to engage in in-depth conversations with patients about vaccines.

*"…obviously more time would be the thing… Someone who might be persuaded, or who wasn't sure. Then it's about it's about time, isn't it?... the conversation about vaccine is often part of a consultation about something else, which has already taken more time than you have."* (P11, Primary Care)

They also felt time was an issue regarding the flu strategy, which was viewed as being released too late to support an impactful communication strategy.

*"Yes, so I would like it to be sooner… So we do, we do have to wait until we get national guidance on eligible populations… So having to wait for that then determines what comms [communications] you are going to do."* (P1, Policy Professional)

Competition and lack of linkage between services was described negatively, as was the ordering process, with impacts on morale and finances.

*"Pharmacies I guess… some of the challenges they [GPs] start to order thousands of flu vaccines and then more pharmacies coming on board to deliver, and then rightly or wrongly there's a competition there, they're competing for the audience, you know."* (P20, Policy Professional)

*"…so you have to order flu like six months in advance. So you're kind of guessing in springtime how many vaccinations you need for winter"* (P2, Policy Professional)

While integration with the community was recognised as important, concerns were raised about resources for practices to do this.

*"…it can require umm trusted people to work closely with those communities and understand what the barrier is for them, and what the concerns are for them and try to address that slowly… I think all of those things require, extra effort… and they require quite considerable resource and that resource would have to be provided… otherwise practices don't have the resource to do that."* (P11, Primary Care)

Some staff described feeling unprepared to tailor information or their approach according to a patient's culture, risk, or perception. Staff found responding to misinformation difficult.

*"I don't think I fully understand why some can and some can't… You know why, why you need porcine free and why this other fella's just told me that he's a Muslim and he can have porcine… but you're telling me you can't? … I don't fully understand the reasoning. If I did, I might be able to talk about it more with them"* (P23, Primary care)

*"I've phone call from a parent really abusive, down the phone asking me, how do I sleep at night?... I can't remember now what she was saying something to do with the brain and swelling and I tried to see whether it's from a reputable source, what she'd reasoned, but there's just no, once they… get into that loop."* (P18, Policy Professional)

## 6. Alliance

**6.1. Current disconnect.** As evidenced above in other themes, there was a disconnect between healthcare and the community. Additionally, poor ethnicity data recording meant that the community was not accurately represented in healthcare data, despite a generally positive perception of the need for data collection:

*"I don't exist as a Polish person on that system. My ethnicity is White Other and the language spoken is English. So I don't exist. In that system, the system, I don't know what the system thinks, but the system doesn't think that I'm Polish. So if this is what we looking at, and this is the data that meant to support the messaging, support the campaigning, and support everything else."* (P10, Community Worker)

**6.2. Co-production and sustainability are key.** People described wanting ongoing relationships with healthcare staff…

*"…years ago they stuck with one doctor you know… but now you see different doctors or different nurses. They don't know what's going on… I need to trust my doctor."* (P45, Public)

…better representation of the communities within healthcare…

*"So actually within healthcare, you know, why would we not have bilingual receptionists? You know, who are recruited from the local, you know, streets and post codes to that practice. That, that would be an ideal way to improve social mobility, improve trust."* (P2, Policy Professional)

…improved flu messaging via working with the community to develop and distribute…

*"…working collectively with community organisations, you know, building, you know, resources, translated materials explaining, I mean this will make, will help and make differences."* (P34, Community worker)

…training to help staff engage effectively with communities and address culturally specific vaccine concerns…

*"Training. I think that is missing here and engagement training. Tips how to work with communities… I see this that is missing unfortunately from people. And mostly with the BME communities… Because um a lot of people doesn't know how to engage BME communities, and they think "ah because I don't know to engage, I will not, will not do anything""* (P13, Community Worker)

…and long-term support of community organisations.

*"…maybe [vaccine] manufacturers should find a way of supporting that third sector, to be then able to support them with getting their intelligence as to what are the barriers, you know, what are the issues for you to be able to come out and do that research on a grassroot level… Those organisations need to be supported, for them to be able to have a long term plans, and to be able to then support the healthcare"* (P10, Community Worker)

## Discussion

### Summary

Influenza outcomes are poorer in ethnically minoritised communities, partly due to low vaccine uptake [6]. This study explored influenza vaccine perceptions among community members, community engagement workers, primary healthcare

staff, and policy professionals. Six inter-related themes emerged. *Beliefs* about vaccine safety, necessity, and efficacy; *Trust* in vaccine information and the healthcare system; *Accessibility* of accurate vaccine information; *Community* opinions/experience; *Healthcare;* and *Alliance.*

## Comparison with existing literature

Several of these findings complement prior work, particularly around misinformation or lack of information, influenza being perceived as low risk, or of people believing alternative medicines or religious behaviours will adequately protect against it, low trust in the healthcare system, concerns about vaccine safety/efficacy, discrimination, and culturally-specific issues (e.g., halal status) [12,18,20]. Where the current study differs, is that while accessibility has been previously highlighted, this has primarily been around physical access such as transport and cost; Nagata, Hernández-Ramos [19], Bhanu, Gopal [21], whereas it was accessibility of information that was most emphasised here. This was found to impact vaccine beliefs and reduce trust due to feelings of low prioritisation. Similar language/literacy barriers were mentioned in the review by Nagata, Hernández-Ramos [19], however these were not major factors in the studies that mentioned them [37–39] with the exception of Schensul, Radda [40], who described a bilingual intervention. In the UK, influenza vaccines are free for eligible individuals, possibly explaining why direct costs are more emphasised in work conducted outside the UK. However, indirect costs such as time off work were mentioned in the current study as physical barriers for some, and have previously been understood as contributing to patients' perceived cost-benefits of vaccination [41]. Those who described these indirect costs, also highlighted the importance of access to flexible vaccine appointments, particularly for people working variable hours, multiple jobs, and who may experience job insecurity, all of which are more likely in ethnically minoritised communities [42].

It is notable that *Trust* was such a key issue in this research as Liverpool has a complex history with regards to ethnicity and diversity, which has affected the way in which people in Liverpool interact with "authority" figures. Liverpool gained much of its wealth through being the largest European transatlantic slave-trading port in the 18th Century [43], and communities in Liverpool have been subject to significant historic and current racism, such as the violent race riots in 1919, primarily against Black residents [44], and the more recent Southport riots in 2024, against Muslims, asylum seekers, and other ethnically minoritised groups [45]. Liverpool is also home to possibly the oldest Black and Chinese communities in Europe, with the former originating partially from African sailors and freed people who had been enslaved [46], and the latter from seamen arriving in the 19th Century [47]. Liverpool also has a substantial Irish population due to migration from Ireland during the Great Hunger of the 1840s [48], during which successful crops were exported from Ireland to England while many Irish people were starving [49]. Additionally, a considerable number of asylum seekers and refugees reside in Liverpool [50]. This indicates a highly unique combination of cultures, values, experiences, and needs, across the individuals who make up the fabric of the city, and historical events which have eroded trust between communities and government.

Indeed, regarding the previous finding that barriers can be specific to communities [22], while the current study identified overlapping barriers across marginalised groups in Liverpool, certain aspects were more prevalent in certain groups. Roma and Romanian individuals were particularly concerned about vaccine safety and trustworthiness of unfamiliar sources, were vulnerable to misinformation due to low literacy and chosen information sources (social media and family members), preferred video/audio information in their languages, perceived religious behaviours as protective, valued free will (feeling distrustful of being told 'what to do'), and feared needles. The Chinese and Hong Kong communities preferred written information in Chinese, perceived alternative medicine as protective, and were wary of phone calls and text messages from unrecognised sources due to concerns about scams. People who had recently arrived in the UK/Liverpool (e.g., asylum seekers) were less sure about how to access the vaccine and feared needles as some had no prior experience of injections. Muslim communities were concerned about halal status of vaccines, and vaccine acceptability in their community related to this. Finally, Polish people were more likely to view influenza vaccination as low priority or

inaccessible due to working long hours or multiple jobs. Some also travelled out of Liverpool to access services in Manchester (which has a Polish speaking healthcare centre) or Poland, where the private healthcare system was described as efficient, responsive, and with long appointments with familiar clinicians – which, alongside poor NHS recording of Polish ethnicity (historically "White Other") indicates their UK vaccine uptake data may be inaccurate. Importantly, the groups described here participated in larger numbers, revealing recurring patterns, but patterns for other groups likely remain unidentified. Interestingly, there were no identified patterns relating to age-groups, which is somewhat surprising due to the vaccine's eligibility status being related to this, and prior research indicating that this is important [24,25].

The findings of this study are consistent with the broader literature on vaccine hesitancy, particularly regarding COVID-19 and MMR vaccines. Common themes such as mistrust in healthcare systems, concerns about vaccine safety and necessity, and the influence of misinformation are prominent with influenza, MMR and COVID-19 vaccine hesitancy. Similar to our participants' uncertainty about influenza vaccine safety and halal status, concerns about COVID-19 vaccine safety, speed of development, and religious acceptability have been well documented [20,22]. Likewise, the MMR vaccine controversy, which is rooted in the discredited claims of an association with autism, shows parallels in how misinformation can persist within communities, particularly when amplified by trusted social networks [51].

Where our findings diverge from COVID-19 and MMR vaccine hesitancy is the centrality of "accessibility of accurate information" rather than physical access. While access issues during the COVID-19 pandemic often focused on vaccine availability, rollout logistics, and structural inequalities [52], in the present study, language and literacy barriers were more common. This distinction highlights how, for influenza vaccination, especially in ethnically minoritised communities, communication gaps may outweigh logistical barriers. Another contrast lies in perceived disease severity. COVID-19 was widely viewed as a high-risk illness, which motivated some to accept vaccination despite hesitancy [18]. In contrast, influenza was often perceived as low risk, echoing findings from MMR research where the disease itself was seen as relatively harmless compared to perceived risks of the vaccine (Brown et al., 2012). This perception of low threat reduces motivation to vaccinate and amplifies the influence of misinformation.

## Strengths and limitations

This study was conducted across a large and diverse sample, including often under-represented groups in research such as Romanian and Roma communities. As one of our participants felt, it can certainly appear that some ethnically minoritised groups have "been marked as the hard to reach, but nobody really wanted to reach them" (P10, Community Worker). This emphasises the importance of working with established community organisations to support recruitment and data collection of marginalised groups, and of not excluding individuals who do not speak English, particularly from conversations about access to healthcare. Additionally, a variety of stakeholders took part in this research, increasing the utility of the findings. Due to the importance of understanding this topic through the lens of the ethnically minoritised groups, the analysis prioritised the viewpoints of members of the public and the community workers, to ensure that the analysis most reflected their experiences. Views of healthcare and policy professionals were used to compare against and complement the views of communities. Finally, while data collection, coding, and analysis was primarily conducted by AP, 14.58% of this was conducted by DC, who reached a similar interpretation of the data before it was integrated into the primary analysis, thus reducing the impact of any preconceptions by AP.

A limitation of this study is that while the focus was to understand a localised population, this does mean that the results may not reflect other geographical areas. However, as the results largely complement prior work, this suggests that the findings may generalise, though specific details will be more applicable to certain communities. Additionally, while many communities did participate, it was not possible to recruit a member of every ethnically minoritised group in Liverpool, or even to speak to multiple members of each group who did take part to establish group-specific patterns. An example of a key group not reflected is the Liverpool Irish Traveller community, who could not be recruited due to time and capacity of the supporting organisations. When recruiting from one community organisation, we extended

our inclusion criteria to include people who were eligible for influenza vaccine or those who had an eligible child. This decision was made because during our engagement with the organisation, they felt that it would reduce trust if we only targeted people who were eligible for influenza vaccine. The groups in this organisation were predominantly women, so the decision was made to extend the criteria to include those women who would be consenting for flu vaccination for their children (N = 8 who were parents, of which N = 4 were also eligible themselves). While this could potentially affect the results as the reasons for consenting to a childhood vaccination may differ to those for an adult vaccination, our interview questions did not focus on childhood vaccination programmes and were centred around the adult seasonal flu vaccination programme, so we do not believe that this will affect the results. Moreover, as noted above, there are some similarities between vaccine hesitance in vaccination for MMR (given predominantly to children in the UK) and influenza, so inclusion of a small number of people with eligible children would not be likely to affect the robustness of our results. Finally, it should be noted that community members were primarily recruited through community organisations, and it is unsurprising that active users of these services trust and value them. A challenge therefore is to consider how to understand perceptions of marginalised individuals who are not involved with such organisations, who may be difficult to access and to plan intervention for.

### Implications for research and/or practice

This study emphasises the need for stakeholders to collaborate in improving access to reliable information through trusted sources (such as local community organisations), while ensuring flexible appointments are easily available and effectively communicated. Communication strategies should be tailored to each community, using the most appropriate language and format for that group, and addressing relevant concerns and misinformation in a culturally appropriate manner. Culturally informed training for healthcare staff would enable them to support this. Other factors contributing to mistrust (e.g., racism and medical scandals) should be effectively and transparently addressed, at both an institutional and interpersonal level. Similarly, stakeholders should foster trust and improve influenza communication by collaborating with and supporting community organisations, particularly regarding their long-term stability. Such support could include funding, up to date/accurate information and support to share this in various formats, training for community workers in how to disseminate influenza information, medical language training for local interpreters, and stability of funded roles. Finally, stakeholders should seek to further encourage ongoing relationships between healthcare and communities, including by improving access to reliable translators, hiring community members, and increasing the consistency of access to familiar healthcare staff. Future research should seek to develop and pilot localised intervention based on these findings. Table 2 may serve as a practical guide to the settings in which specific barriers occur, thereby supporting the design of tailored intervention strategies. Researchers should also consider how to access and support marginalised individuals not affiliated with a supporting organisation.

### Conclusion

The current study found that influenza vaccine hesitance in ethnically minoritised communities in Liverpool stems from *Beliefs* about the vaccine being unsafe, unnecessary, and/or ineffective, low *Trust* in the healthcare system, poor *Accessibility* of accurate vaccine information, *Community* perceptions and influence, *Healthcare* staff having low motivation and preparedness to address cultural concerns, and a currently unsatisfactory *Alliance* between healthcare and the local communities. Stakeholders should address influenza vaccine hesitance via collaborating to improve access to reliable and relevant information, increase trust by hiring community members and ensuring consistent access to familiar staff, and foster alliance by long-term support of community organisations. Future research should use these insights to develop, pilot, and evaluate localised and targeted intervention, and consider how to improve influenza outcomes in individuals not affiliated with community organisations who appear to be key access points for both research recruitment and behaviour change intervention.



## Supporting information

**S1 File. Interview and focus group topic guides for each participant group.**
(PDF)

## Acknowledgments

We extend our gratitude to the healthcare and local authority networks, and community organisations that supported the recruitment process for this study. Special thanks go to the community organisations that provided access, interpretation of participant-facing materials, and translation services, to enable a diverse group of participants to take part in this research. We would also like to thank Claire Adshead for her help with data transcription.

## Author contributions

**Conceptualization:** Marie Claire Van Hout, Catharine Montgomery.

**Data curation:** Anna Powell, Deborah Connors, Catharine Montgomery.

**Formal analysis:** Anna Powell, Deborah Connors.

**Funding acquisition:** Marie Claire Van Hout, Catharine Montgomery.

**Investigation:** Anna Powell, Deborah Connors.

**Methodology:** Anna Powell, Marie Claire Van Hout, Deborah Connors, Catharine Montgomery.

**Project administration:** Anna Powell, Deborah Connors, Catharine Montgomery.

**Supervision:** Anna Powell, Catharine Montgomery.

**Validation:** Deborah Connors.

**Visualization:** Anna Powell, Deborah Connors.

**Writing – original draft:** Anna Powell.

**Writing – review & editing:** Anna Powell, Marie Claire Van Hout, Deborah Connors, Catharine Montgomery.

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
