## [Decision Letter · Decision Letter 0]

22 Jun 2025

Dear Dr. Powell,

Thank you for submitting your manuscript to PLOS ONE. After careful consideration, we feel that it has merit but does not fully meet PLOS ONE’s publication criteria as it currently stands. Therefore, we invite you to submit a revised version of the manuscript that addresses the points raised during the review process.

We look forward to receiving your revised manuscript.

Kind regards,

Ilhem Berrou, PhD

Academic Editor

PLOS ONE

Journal Requirements:

2.  Thank you for stating the following financial disclosure: [This study was funded by The Pandemic Institute, through partnership with CSL Seqirus UK Limited [TPI-SEQ3-CM].]. 

4. We note that Supporting Information S1 Appendix.pdf in your submission contain copyrighted images.

All PLOS content is published under the Creative Commons Attribution License (CC BY 4.0), which means that the manuscript, images, and Supporting Information files will be freely available online, and any third party is permitted to access, download, copy, distribute, and use these materials in any way, even commercially, with proper attribution. For more information, see our copyright guidelines: http://journals.plos.org/plosone/s/licenses-and-copyright.

1. You may seek permission from the original copyright holder of Supporting Information S1 Appendix.pdf to publish the content specifically under the CC BY 4.0 license.

5. Please include captions for your Supporting Information files at the end of your manuscript, and update any in-text citations to match accordingly. Please see our Supporting Information guidelines for more information: http://journals.plos.org/plosone/s/supporting-information .

Additional Editor Comments:

Thank you for submitting this well written manuscript.

In addition to the comments raised by reviewers, please also address the following:

Introduction

Provide relevant data on vaccine uptake among racially minoritised communities in UK and a comparison with global patterns.

Provide information to the international reader about how influenza vaccine is offered/ can be accessed in UK (or just England).

You provide data on Liverpool population which is the context of this paper. It will be helpful to compare vaccine uptake in other cities e.g. London or Manchester other others with similar ethnic mix.

It would be helpful to theme the barriers to vaccine uptake to ensure clarity on the role/ impact of those barriers. Some barriers relate to the service and how it is offered, others relate to the individual, community or the relationship certain communities have with the local system e.g. historical links to slavery, racism etc...

Methods

Please justify including adult (aged 18+) members of the public who self-identified as being part of an ethnically minoritised group and eligible for a seasonal influenza vaccine (or had an eligible child). Childhood vaccination via schools may be different to adult flu vaccination service, and the considerations will be different. This difference should also be addressed in the results/ analysis.

Results and Discussion

The figure summarising the themes is great. Could the authors consider mapping the themes and the various barriers identified the influenza vaccine programme highlighting where these barriers may rise and may be addressed.

Reviewers' comments:

Reviewer's Responses to Questions

**Comments to the Author**

1. Is the manuscript technically sound, and do the data support the conclusions?

Reviewer #1: Yes

2. Has the statistical analysis been performed appropriately and rigorously?

Reviewer #1: N/A

3. Have the authors made all data underlying the findings in their manuscript fully available?

Reviewer #1: No

4. Is the manuscript presented in an intelligible fashion and written in standard English?

Reviewer #1: Yes

Reviewer #1: Thank you for submitting a good and timely article. The topic is highly relevant, and the manuscript has the potential to make a valuable contribution to the literature. With a few revisions to improve clarity and depth in key sections, this work would be well-positioned for publication.

Introduction:

The introduction could be strengthened to provide a more compelling rationale for the study. For instance, in lines 40–41, global influenza-related mortality is mentioned, but no reference is made to UK-specific data. While global statistics offer important context, they may not directly support the case for the study’s relevance within the UK healthcare system. Consider rephrasing this section to better reflect the local context.

In line 42, the statement regarding the reduction in mortality and morbidity through vaccination would be more impactful if supported by specific figures. Additionally, incorporating UK-specific data—especially broken down by ethnicity—would enhance the rationale for the study. For example, how do influenza outcomes differ between ethnic minority groups and the White British population? Quantifying this disparity could help justify the focus and scope of the research.

Methods:

The Methods section would benefit from greater clarity. The study design should be explicitly stated (e.g., qualitative, quantitative, or mixed-methods). More information is needed on the sample size—specifically how it was calculated or justified. It would also be helpful to describe the sampling strategy in more detail, including how participants were selected and whether any bias may have been introduced.

Furthermore, please include information about who conducted the interviews and what training or experience they had in qualitative research, as this can impact the quality and reliability of the data collected. Providing the interview schedule or guide would also strengthen transparency. Additionally, clarifying how many interviews were conducted individually versus in focus groups would improve understanding of the data collection approach.

Results and Discussion:

In the Results section, consider clearly identifying the most commonly reported themes to help readers grasp the key findings more readily. It would also be valuable to explore whether participants of different age groups expressed distinct concerns or priorities, as this could inform age-specific public health messaging.

In the Discussion, it would be helpful to compare your findings with those from studies on other ethnic minority groups, both within the UK and internationally. If there are findings that appear unique to the Liverpool context, please consider discussing the local factors that might explain these. This will help readers understand how widely the findings might apply beyond the study setting.

Additionally, drawing parallels with vaccine hesitancy related to other vaccines—such as COVID-19 or MMR—could add depth and context to your interpretation. If a particular theme emerged as especially dominant, it would be useful to explore its implications in greater detail, particularly in relation to future intervention strategies or research directions.

**Do you want your identity to be public for this peer review?** For information about this choice, including consent withdrawal, please see our Privacy Policy

Reviewer #1: No

---

## [Author Response · Author response to Decision Letter 1]

19 Aug 2025

Dear Professor Chenette,

Thank you for giving us the opportunity to submit a revised draft of our manuscript titled “A thematic analysis of flu vaccine hesitance in ethnically minoritised communities in Liverpool” to PLOS ONE (Manuscript ID PONE-D-25-20640). We appreciate the time and effort that the Editor and the Reviewer have dedicated to providing valuable feedback on our manuscript. We are grateful for the insightful comments on our paper. We have made changes to the existing manuscript to respond to the Editor and the Reviewer’s comments, and we believe that this has significantly strengthened our manuscript. We hope that you now find this acceptable for publication in PLOS ONE.

As requested as an addition to the cover letter, we would like to clarify that the statement: “The funders had no role in study design, data collection and analysis, decision to publish, or preparation of the manuscript.” is correct. Please can this be amended on the online submission.

We look forward to hearing from you in due time regarding our submission and to respond to any further questions and comments you may have.

Sincerely,

Anna Powell

Regarding the Editor’s specific comments:

Response: We have amended our manuscript and other files to match PLOS ONE’s style requirements.

2. Thank you for stating the following financial disclosure: [This study was funded by The Pandemic Institute, through partnership with CSL Seqirus UK Limited [TPI-SEQ3-CM].]. Please state what role the funders took in the study. If the funders had no role, please state: ""The funders had no role in study design, data collection and analysis, decision to publish, or preparation of the manuscript."" If this statement is not correct you must amend it as needed. Please include this amended Role of Funder statement in your cover letter; we will change the online submission form on your behalf.

Response: The statement: “The funders had no role in study design, data collection and analysis, decision to publish, or preparation of the manuscript.” is correct, please can this be amended on the online submission.

3. We note that you have indicated that there are restrictions to data sharing for this study. For studies involving human research participant data or other sensitive data, we encourage authors to share de-identified or anonymized data. However, when data cannot be publicly shared for ethical reasons, we allow authors to make their data sets available upon request. Before we proceed with your manuscript, please address the following prompts: a) If there are ethical or legal restrictions on sharing a de-identified data set, please explain them in detail (e.g., data contain potentially identifying or sensitive patient information, data are owned by a third-party organization, etc.) and who has imposed them (e.g., a Research Ethics Committee or Institutional Review Board, etc.). Please also provide contact information for a data access committee, ethics committee, or other institutional body to which data requests may be sent. b) If there are no restrictions, please upload the minimal anonymized data set necessary to replicate your study findings to a stable, public repository and provide us with the relevant URLs, DOIs, or accession numbers. Please update your Data Availability statement in the submission form accordingly.

Response: Our Institutional Ethics Research Committee (LJMUREC) approved the study on the basis that we will deposit anonymised data on our open access data repository opendata.ljmu.ac.uk, on a restricted basis. If someone wishes to access the data via the repository, they click a link on the web page which sends the request to the corresponding author through the data access team. Access requests will be reviewed within 5 working days and assuming that the requests are reasonable (e.g. academic or health care researchers working in this field), it will be granted. We have now included this URL as a placeholder in our revision (the final URL will be available after manuscript acceptance), and will update the data repository availability statement in the submission form.

4. We note that Supporting Information S1 Appendix.pdf in your submission contain copyrighted images. In the figure caption of the copyrighted figure, please include the following text: “Reprinted from [ref] under a CC BY license, with permission from [name of publisher], original copyright [original copyright year].”

Response: We have removed these images from the Supporting Information S1 File.

Response: We have included the correct style captions for any supplementary materials and have amended the text accordingly.

6. Please review your reference list to ensure that it is complete and correct. If you have cited papers that have been retracted, please include the rationale for doing so in the manuscript text, or remove these references and replace them with relevant current references. Any changes to the reference list should be mentioned in the rebuttal letter that accompanies your revised manuscript. If you need to cite a retracted article, indicate the article’s retracted status in the References list and also include a citation and full reference.

Response: Thank you for bringing this to our attention. We have checked our reference list, and we noticed that a CDC reference had been moved to a new URL on their website (citation number 10). We have updated to reflect this. Our reference list now contains no redacted or redirected papers or links.

7. Provide relevant data on vaccine uptake among racially minoritised communities in UK and a comparison with global patterns.

Response: We have provided a summary of trends in flu vaccine uptake data and ethnicity for the 2024-25 flu season in the UK and added some information on the global context.

8. Provide information to the international reader about how influenza vaccine is offered/ can be accessed in UK (or just England).

Response: We have added this information at the end of paragraph 1 in the introduction.

9. You provide data on Liverpool population which is the context of this paper. It will be helpful to compare vaccine uptake in other cities e.g. London or Manchester other others with similar ethnic mix.

Response: We have added a comparison to inequalities in flu vaccine uptake in Manchester, another city in the North of England with comparable ethnic mix. We have included a brief mention of vaccine rates in London, but we believe this may not be as relevant as London is a much larger and more diverse city and will be subject to similar but unique barriers and inequalities due to its diversity and size, and its location in the South of England.

10. Method: Please justify including adult (aged 18+) members of the public who self-identified as being part of an ethnically minoritised group and eligible for a seasonal influenza vaccine (or had an eligible child). Childhood vaccination via schools may be different to adult flu vaccination service, and the considerations will be different. This difference should also be addressed in the results/ analysis.

Response: The editor rightfully notes the differences between the childhood and adult vaccination programmes for influenza vaccination and that this could affect the results. From one community organisation, we included those who had eligible children too because during our engagement with the organisation, they felt that it would reduce trust if we only targeted people who were eligible and they may feel that we were pushing vaccination on them. This particular organisation had a population of predominantly women attending its group sessions, so the decision was made to extend the criteria as those women would be consenting for flu vaccination for their children (N = 8 who were parents, of which N = 4 were also eligible themselves). Our interview questions did not specifically focus on childhood vaccination programmes and were centred around the adult seasonal flu vaccination programme, so we do not believe that this will affect the results. However, we have noted this as a limitation in the discussion section.

11. Results and Discussion: The figure summarising the themes is great. Could the authors consider mapping the themes and the various barriers identified the influenza vaccine programme highlighting where these barriers may rise and may be addressed.

Response: Thank you for the positive feedback and for this great idea. We have added the described table in the Results section (Table 2). We have also added a sentence in the Implications for research and/or practice section highlighting that this Table serves as a guide to the locations in which specific barriers occur, and so may be used to support the development of tailored intervention.

Regarding the Reviewer’s specific comments:

12. Introduction: The introduction could be strengthened to provide a more compelling rationale for the study. For instance, in lines 40–41, global influenza-related mortality is mentioned, but no reference is made to UK-specific data. While global statistics offer important context, they may not directly support the case for the study’s relevance within the UK healthcare system. Consider rephrasing this section to better reflect the local context.

Response: Thank you to Reviewer 1 for this helpful comment. We have rephrased this section and have also added more information on the local (Liverpool) and UK context to allow a reader to see the scale of the problem in the UK. We have also added more text at the request of the editor on variations in flu vaccine uptake for different ethnicities in the UK and globally.

13. In line 42, the statement regarding the reduction in mortality and morbidity through vaccination would be more impactful if supported by specific figures. Additionally, incorporating UK-specific data—especially broken down by ethnicity—would enhance the rationale for the study. For example, how do influenza outcomes differ between ethnic minority groups and the White British population? Quantifying this disparity could help justify the focus and scope of the research.

Response: We have amended this section and rewritten in line with comments from Reviewer 1 and the Editor. We now included data on mortality across the UK and in Liverpool, and national data on racial inequalities in emergency hospital admissions.

14. Methods: The Methods section would benefit from greater clarity. The study design should be explicitly stated (e.g., qualitative, quantitative, or mixed-methods). More information is needed on the sample size—specifically how it was calculated or justified. It would also be helpful to describe the sampling strategy in more detail, including how participants were selected and whether any bias may have been introduced.

Response: We agree and have added a sentence specifying a qualitative study design. The section on recruitment has also been rewritten to clarify how participants were identified, and to acknowledge that “the use of community organisations as gatekeepers to recruit members of the public likely introduced a positive bias towards these organisations.” Finally, we have described how information power was used to inform sample size.

15. Furthermore, please include information about who conducted the interviews and what training or experience they had in qualitative research, as this can impact the quality and reliability of the data collected. Providing the interview schedule or guide would also strengthen transparency. Additionally, clarifying how many interviews were conducted individually versus in focus groups would improve understanding of the data collection approach.

Response: Thank you for this comment. We have added detail on the researchers’ experience in the Procedure section. The topic guides are included in the S1 file, but this has been made clearer in the paper (again in the Procedure section). We have also clarified near the end of the Participants and Recruitment section that “Focus groups were conducted across 34 participants (with individuals from the same participant group), and consisted of between two and five people, while 62 individual interviews were conducted.”

16. Results: In the Results section, consider clearly identifying the most commonly reported themes to help readers grasp the key findings more readily. It would also be valuable to explore whether participants of different age groups expressed distinct concerns or priorities, as this could inform age-specific public health messaging.

Response: We agree that this could have been clearer, and have added information on this at the start of the Results section. Specifically, we have highlighted that the themes Beliefs, Trust, and Accessibility were the most prominent, and we have indicated which subthemes within these were most prevalent. We have also clarified that the results are presented in an order that best supports a coherent and meaningful narrative, rather than reflecting frequency.

We did not notice any patterns across or between age groups, which is somewhat surprising as the vaccine’s eligibility status is linked to age, and as age has been identified as important in prior literature. We have put a sentence clarifying this in the paper under “Comparison with existing literature”.

17. In the Discussion, it would be helpful to compare your findings with those from studies on other ethnic minority groups, both within the UK and internationally. If there are findings that appear unique to the Liverpool context, please consider discussing the local factors that might explain these. This will help readers understand how widely the findings might apply beyond the study setting.

Additionally, drawing parallels with vaccine hesitancy related to other vaccines—such as COVID-19 or MMR—could add depth and context to your interpretation. If a particular theme emerged as especially dominant, it would be useful to explore its implications in greater detail, particularly in relation to future intervention strategies or research directions.

Response: The discussion is now structured to allow readers to see comparisons to other ethnic groups and factors that are unique to Liverpool. Reviewer 1 raises an interesting point regarding hesitancy towards other vaccines. While there are some similar themes relates to hesitancy towards COVID-19 and MMR vaccines, those two vaccine programmes have distinct factors which we did not identify. Because COVID-19 vaccines are novel there is a much larger weight given to concerns about vaccine safety and development, whereas for MMR, there are still remaining concerns regarding links to autism despite retraction of the erroneous Wakefield paper. There are similarities in terms of trust and misinformation which we have attempted to highlight with the overall implications for future intervention strategies being that community-based assets are key. We have included a brief discussion of the similarities and differences in the discussion.

---

## [Editor Report · Decision Letter 1]

16 Sep 2025

A thematic analysis of flu vaccine hesitance in ethnically minoritised communities in Liverpool

PONE-D-25-20640R1

Dear Dr. Powell,

We’re pleased to inform you that your manuscript has been judged scientifically suitable for publication and will be formally accepted for publication once it meets all outstanding technical requirements.

Kind regards,

Ilhem Berrou, PhD

Academic Editor

PLOS ONE

---

## [Editor Report · Acceptance letter]

PONE-D-25-20640R1

PLOS ONE

Dear Dr. Powell,

I'm pleased to inform you that your manuscript has been deemed suitable for publication in PLOS ONE. Congratulations! Your manuscript is now being handed over to our production team.

Kind regards,

on behalf of

Dr. Ilhem Berrou

Academic Editor

PLOS ONE